# Percutaneous kyphoplasty for osteoporotic vertebral compression fractures improves spino-pelvic alignment and global sagittal balance maximally in the thoracolumbar region

Zhong Cao[1], Guodong Wang[2], Wenpeng Hui[1], Bo Liu[1], Zhiyong Liu[1], Jianmin Sun[2]*

1 Department of Orthopaedic Surgery, Shandong Provincial ENT Hospital affiliated to Shandong University, Shandong Provincial ENT Hospital, Jinan, P.R. China, 2 Department of Spine Surgery, Shandong Provincial Hospital affiliated to Shandong University, Jinan, P.R. China

* sunjmsunjmsd@163.com

**Data Availability Statement:** All relevant data are within the manuscript and its Supporting Information files.

## Abstract

### Background

Osteoporotic vertebral compression fractures (OVCFs) often cause local kyphosis. Percutaneous kyphoplasty (PKP) is a common method for the treatment of local kyphosis. However, the influence of kyphoplasty on spino-pelvic alignment and global sagittal balance when performed at specific treatment sites in the spine remains unclear.

The purpose of the study is to investigate the influence of different fracture sites and PKP treatment on the spino-pelvic alignment and global sagittal balance in patients with OVCFs.

### Methods

90 patients with OVCF who underwent PKP were included in the retrospective study. According to the site of the fractured vertebrae, all the cases were divided into 3 groups: Main thoracic (MT) group (T1 to T9), Thoracolumbar (TL) group (T10 to L2) and Lumbar (LU) group (L3 to L5). 26 healthy elderly volunteers (aged over 59) were enrolled as the control group. Sagittal spino-pelvic parameters were measured on the full-spine radiographs preoperatively and postoperatively. Information of sagittal spino-pelvic parameters and global sagittal balance was gathered.

### Results

Compared with the Control group, TL group showed significant differences in almost all parameters, except pelvic incidence (PI) and lumbar lordosis (LL). While only local sagittal parameters (Thoracic kyphosis (TK), Thoracolumbar kyphosis (TLK), LL) were significantly different in MT group. There was no significant difference in almost all of the parameters except for PT and TPA in LU group.

**Funding:** The authors received no specific funding for this work.

**Competing interests:** The authors have declared that no competing interests exist.

**Abbreviations:** OVCF, Osteoporotic vertebral compression fracture; PKP, Percutaneous kyphoplasty; MT, Main thoracic; TL, Thoracolumbar; LU, Lumbar; VAS, Visual analogue scale; PI, Pelvic incidence; PT, Pelvic tilt; SS, Sacral slope; TK, Thoracic kyphosis; TLK, Thoracolumbar kyphosis; LL, Lumbar lordosis; SSA, Spinosacral angle; TPA, T1 pelvic angle; SVA, Sagittal vertical axis.

Correspondingly, the sagittal parameters of TL group improved best after PKP, except for thoracic kyphosis (TK) and sagittal vertical axis (SVA). In MT group, only TLK was significantly decreased, while in LU group, only local kyphosis Cobb angle and SSA were improved.

## Conclusions

OVCF mainly occurs in the thoracolumbar region. Compared with MT group and LU group, OVCF occurred in the thoracolumbar region had greater influence on the spino-pelvic alignment and global sagittal balance. When PKP was performed, the improvement of sagittal balance parameters of TL group was the best in the three groups.

## Introduction

Sagittal balance is a state in which an individual maintains a stable standing position with minimal muscle effort [1]. This state is essential for maintaining normal spinal biomechanics. Several spinal diseases can cause sagittal imbalance, such as spinal deformities, spinal degenerative diseases etc.[2–4] Most researchers are more concerned with sagittal imbalances caused by spinal deformity and degeneration, while sagittal imbalance caused by OVCFs has received less attention [2].

Zhang YL et al confirmed that OVCFs can change the local sagittal alignment of the spine and multiple vertebral compression fractures can even lead to sagittal imbalance [2, 5]. Among patients with OVCFs, the incidence of thoracolumbar vertebral fracture is the most common due to the special anatomical structure and biomechanical characteristics of the thoracolumbar spine itself [6, 7]. Whether there is a difference in the effect of thoracolumbar fracture site on sagittal balance has not yet been studied. Moreover, differences in sagittal balance improvement after PKP procedure in different fracture sites have not been reported.

Our study retrospectively analyzed the sagittal balance parameters of 90 patients with OVCFs treated with PKP and selected 26 healthy elderly volunteers as the control group. We tried to analyze the differences in sagittal balance parameters after a vertebral fracture at different sites and to analyze the differences in sagittal balance improvement after PKP at different fracture sites.

## Materials and methods

A total of 90 patients with OVCFs receiving PKP treatment between January 2013 and July 2018 in Shandong Provincial Hospital affiliated to Shandong University were enrolled. Three senior spine surgeons from the same surgical group operated on all patients. The patient-related data and imaging materials were obtained from the electronic medical record management system of Shandong Provincial Hospital affiliated to Shandong University. The study has been approved by the Ethics Committee of Shandong Provincial Hospital affiliated to Shandong University.

### Inclusion criteria [2, 8]

(1).The vertebral compression ratio of the injured vertebrae was less than 80%; (2).Osteoporosis was confirmed via bone mineral density in elderly patients; (3).All fractured vertebrae showed a high signal intensity on short T1 inversion recovery(STIR) magnetic resonance (MR) images and a low signal intensity on T1-weighted MR images; (4).The imaging data

were complete, including the preoperative and postoperative follow-up standing X-ray films of the whole spine with pelvis and femoral heads, and three-dimensional CT and MRI of the thoracolumbar spine.

## Exclusion criteria [2, 8]

(1).Patients with lumbar disc herniation, spondylolisthesis, scoliosis, spinal osteoarthritis, ankylosing spondylitis, spinal tumors and spinal tuberculosis; (2).Patients with a history of spinal surgery; (3).Patients with hip and knee joint limitations (a history of hip and knee joint diseases, or abnormal hip and knee joint mobility in the medical records); (4).Patients with spinal cord compression with clinical manifestations of spinal cord and cauda equina nerve injury; (5).Patients with pathogenic fracture caused by a tumor or incomplete posterior wall of the vertebral body; (6).Patients who could not stand upright independently or who did not obtain a standing X-ray film.

After enrollment, the medical history of each patient was reviewed. The number of spinal vertebral fractures and the locations of the fractures were recorded. The demographic data and radiographic findings including plain radiography, computerized tomography, and MR imaging were recorded. The visual analogue scale (VAS) was assessed preoperatively and postoperatively. Full-length radiographs were analyzed for spino-pelvic sagittal parameters.

The sagittal balance of the patient was analyzed by standing radiographs of the whole spine, including the pelvis and the femoral heads[9–17] The fingers of the patient were resting on the clavicles, a position described as reproducible and reliable[17–19].

The sagittal parameters are greatly affected by the standing posture. Based on the numerous studies published by scholars, we chose the following positions for photography [20–23].

1. A natural standing lateral position.

2. Eyes were looking straight ahead.

3. The sagittal plane of the torso was perpendicular to the tube.

4. The hip and knee joints were as straight as possible, and the feet were spaced shoulder width apart.

5. The elbows were in flexion, wrists were in flexion, hands were clenched, and the fingers were resting on the clavicles.

## Radiographic analysis[24]

The standing lateral radiographs were obtained preoperatively and within 2–3 days postoperatively. The main radiological parameters for measuring the sagittal alignment were as follows: PI, pelvic tilt (PT), sacral slope (SS), local kyphosis Cobb angle, TK, TLK, LL, PI-LL, SVA, spino-sacral angle (SSA), and T1 pelvic angle (TPA). The data of these measured parameters were recorded by two investigators using Surgimap software (version: 2.2.14.1, Nemaris, Inc., New York, NY, USA).

The spino-pelvic sagittal parameters are described in Figs 1–6[20, 24–27].

According to the site of the fractured vertebrae, all patients were divided into 3 groups: the MT group (the fractured vertebrae were located between T1 and T9), TL group (the fractured vertebrae were located between T10 and L2) and LU group (the fractured vertebrae were located between L3 and L5). The improvement in the spino-pelvic sagittal parameters before and after the operation was calculated, and the differences in the sagittal parameters among the groups were compared.

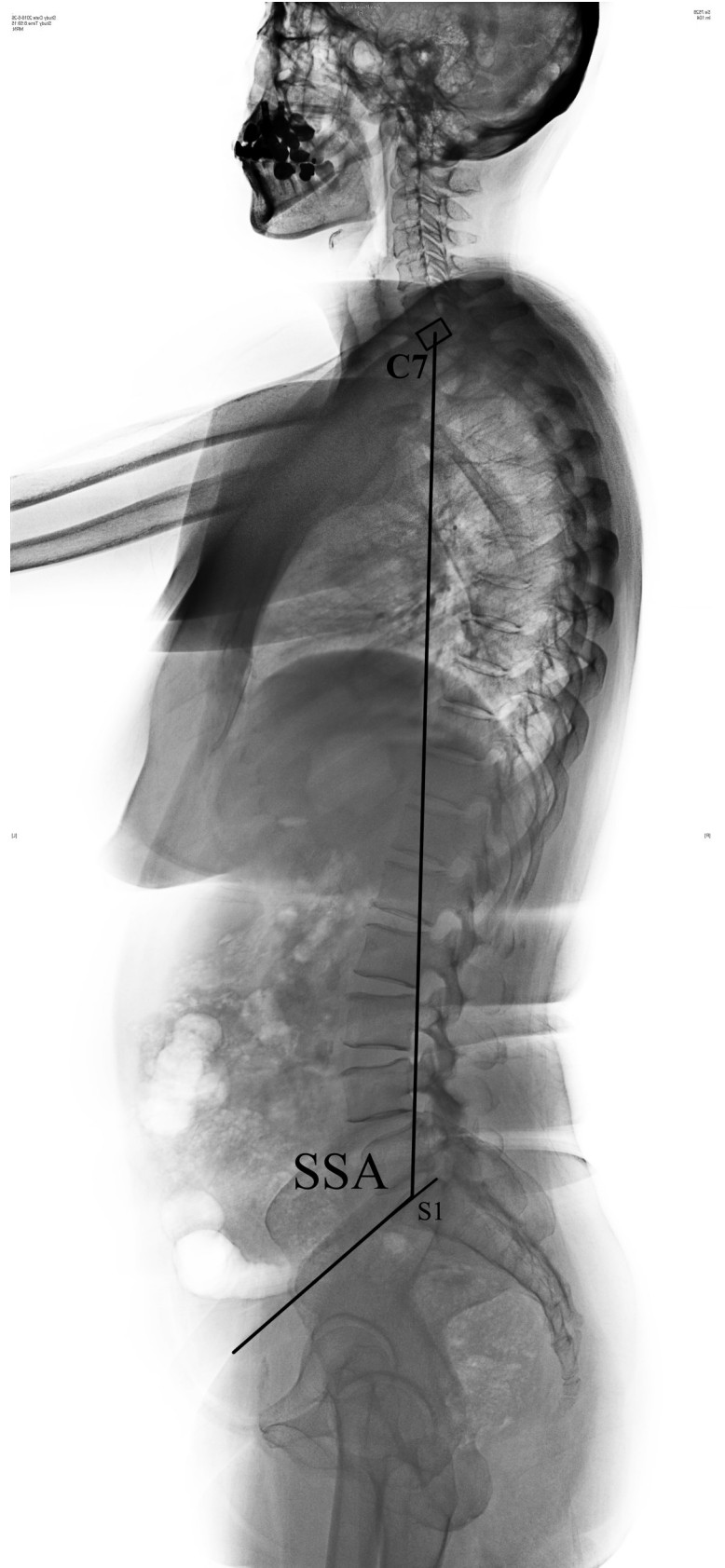

**Fig 1. Radiographic measurements of SSA.** The angle between the upper endplate of the S1 and the line connecting the midpoint of the C7 to the midpoint of the upper endplate of the S1.

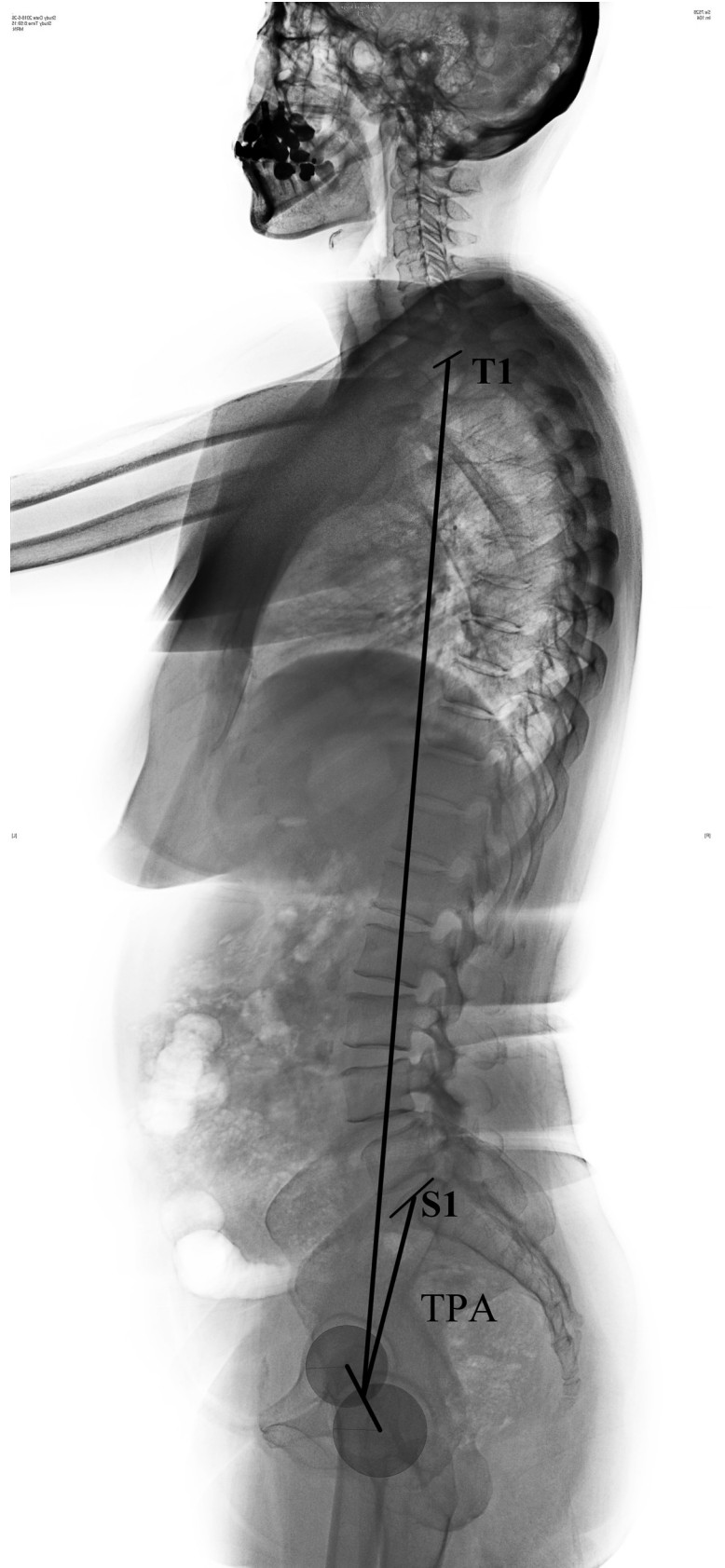

**Fig 2. Radiographic measurements of TPA.** The angle between the midpoint of the upper endplate of T1 and S1 to the midpoint of the femoral head.

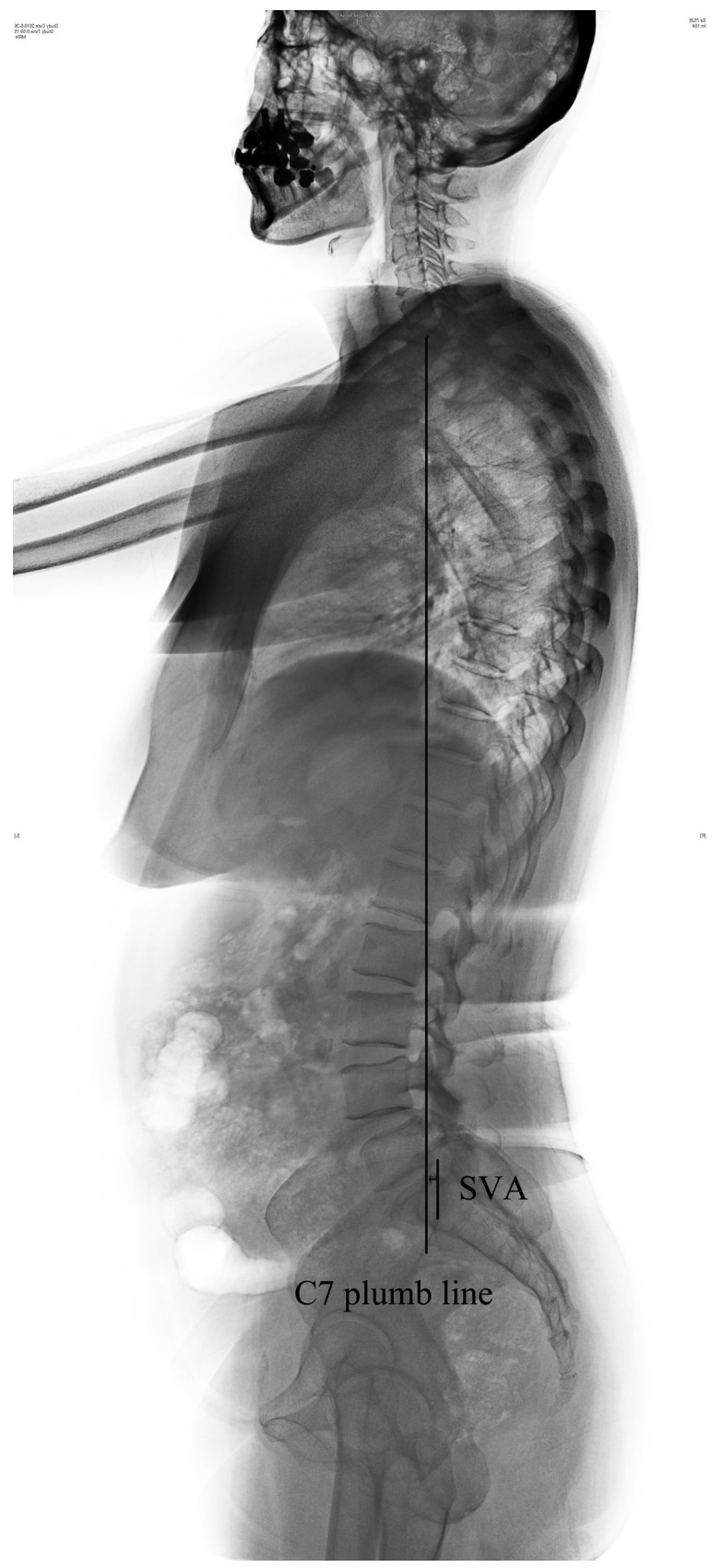

**Fig 3. Radiographic measurements of SVA.**

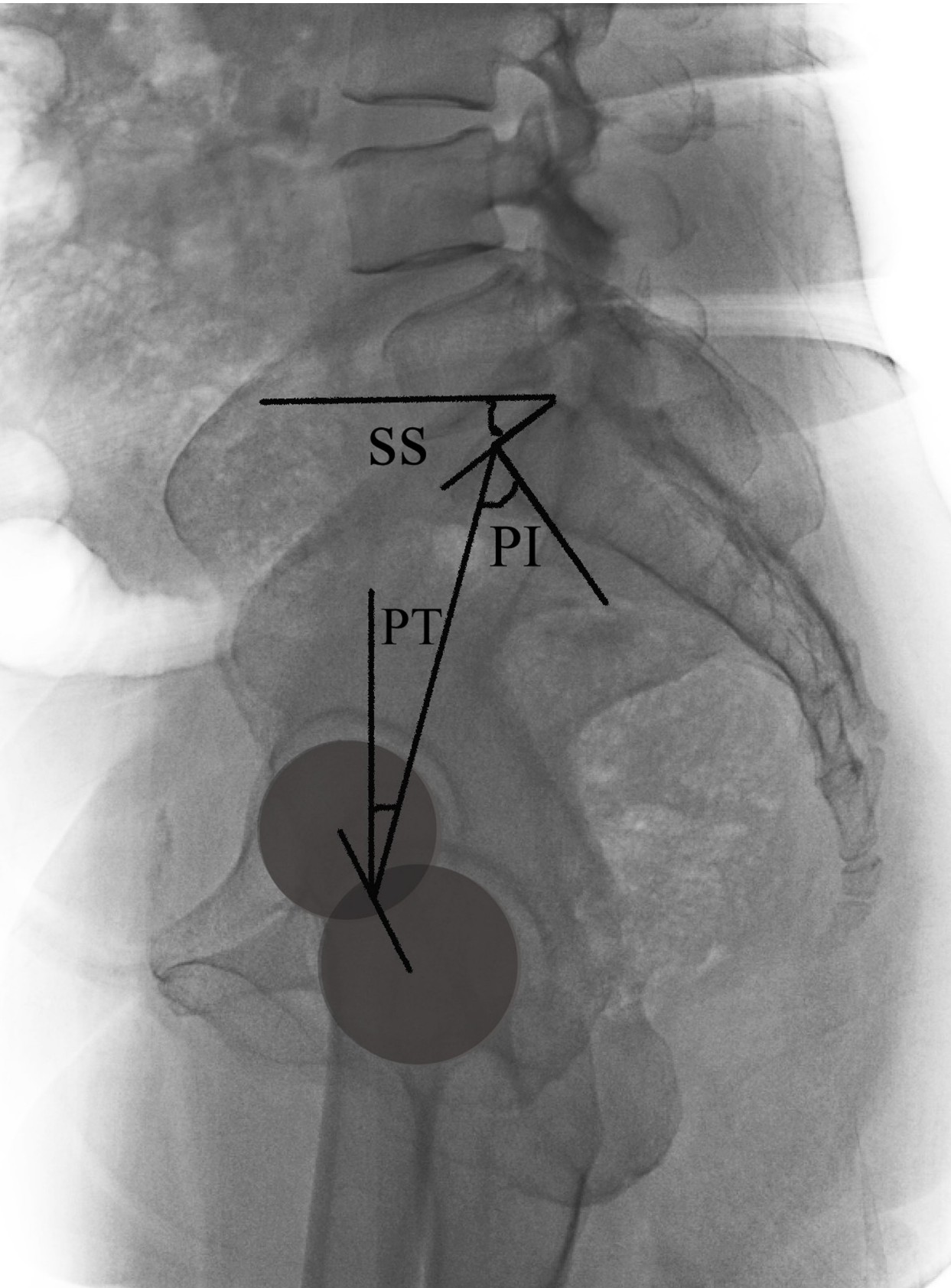

**Fig 4. Radiographic measurements of pelvic sagittal parameters (PI, PT and SS).**

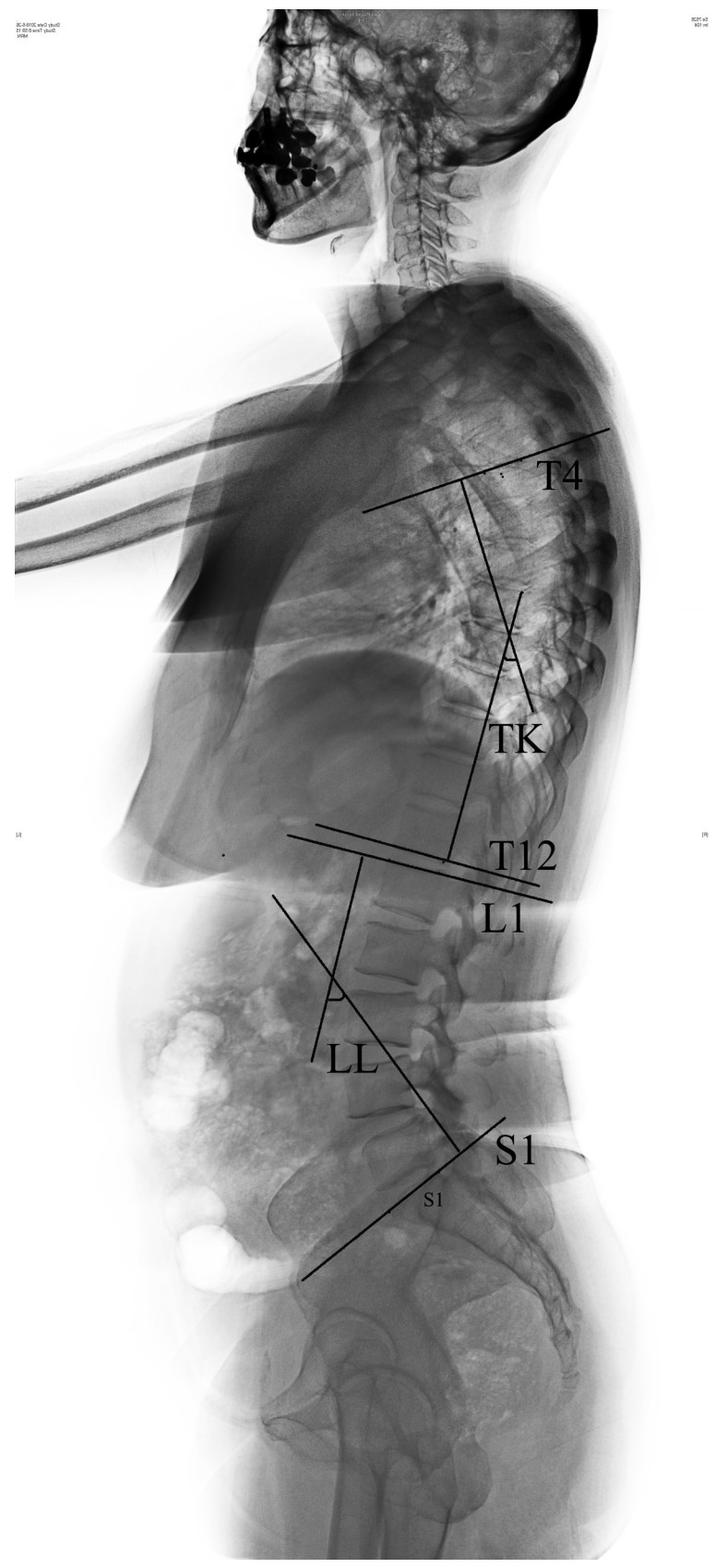

**Fig 5. Radiographic measurements of spinal sagittal parameters (TK, LL).**

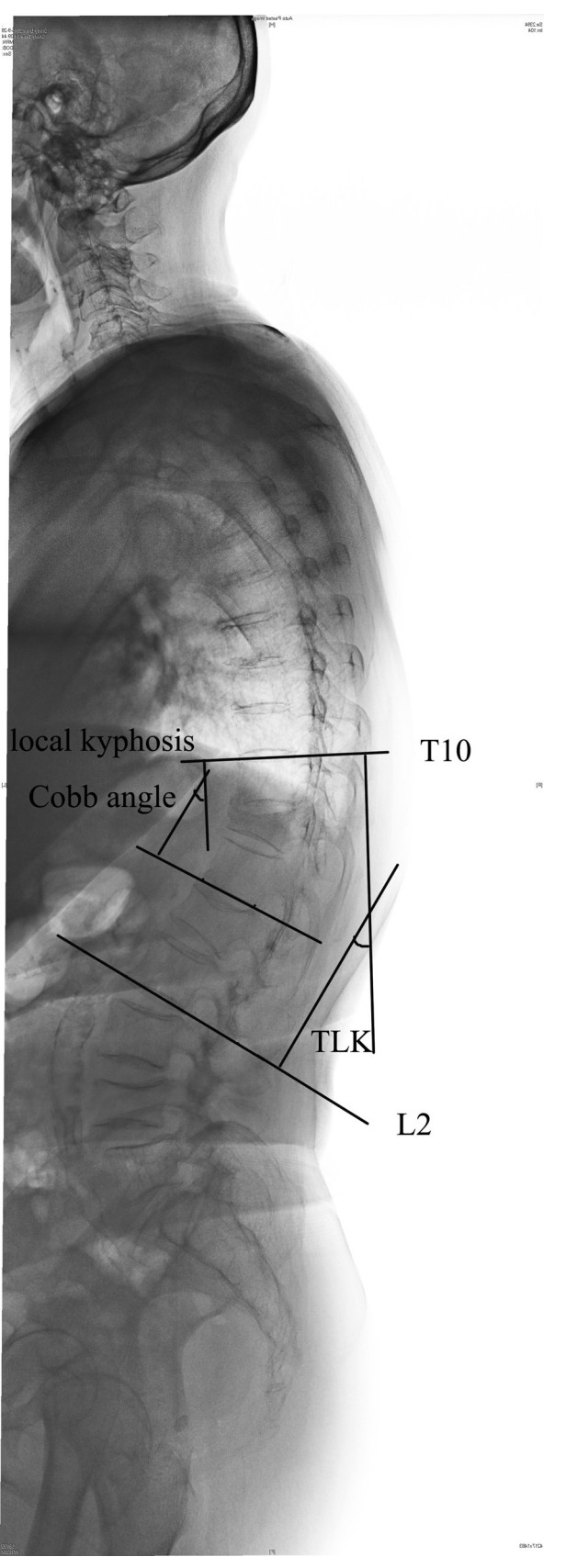

**Fig 6. Radiographic measurements of spinal sagittal parameters (TLK, local kyphosis Cobb angle).** Local kyphosis Cobb angle: the angle formed by the upper endplate of the vertebra above the fractured vertebra and the lower endplate of the vertebra below the fractured vertebra.

Furthermore, all parameters were compared with those of 26 healthy elderly volunteers. Their ages range from 59 to 79 years. The control group consisted of healthy elderly volunteers of a similar age who underwent physical examinations at the authors' hospital and did not have obvious back pain or a history of osteoporotic fracture of the spine.

## Statistical analysis

All statistical analyses were conducted using the IBM Statistics Package for Social Sciences (IBM SPSS Statistics 24, SPSS Inc., Chicago, IL, USA). The normality of the data was evaluated using the Shapiro–Wilk test. Data were expressed as the means ± standard deviations. For non-normally distributed variables, we used medians and interquartile ranges. The independent sample t test was used to compare the data between two groups. If the data did not meet a normal distribution, the comparison was performed using the Mann-Whitney U test. The nonparametric test (K-W test) was used to compare the data among multiple groups. The mean values and standard deviations of the preoperative and postoperative radiological parameters were determined, and changes were evaluated using a paired-sample t test. If the changes did not meet the normal distribution, the comparison was performed using Wilcoxon Signed Ranks test. All statistical tests were two tailed, and a p-value $<0.05$ was considered statistically significant. The correlations between parameters were analyzed by Pearson correlation analysis. Moreover, the power of the study was estimated using a post-hoc analysis with G*Power software (version 3.1.9.4, Franz Faul, Universität Kiel, Germany).

## Results

### Patient data

Ninety patients met the inclusion criteria. They were 70 females and 20 males with a mean age of 69.3±8.1years. The total number of fractured vertebrae was 124. There was no significant differences in age (p = 0.654) and VAS score (p = 0.840 preoperatively; p = 0.352 postoperatively) among the three groups. The VAS scores were significantly decreased after the operation (p < 0.05). Demographic characteristics of the three groups are summarized in Table 1.

### Preoperative radiological measures

Almost all of the preoperative parameters of the TL group were significantly different from those of the healthy volunteers, except for the PI and LL. The TK, TLK, PT, SVA, and TPA were all significantly larger and the SS and SSA were significantly smaller than those of the

**Table 1. Comparison of demographic characteristics of the three groups stratified according to the site of OVCFs.**

|  | MT group | TL group | LU group | P-value |
|---|---|---|---|---|
| Number of patients | 9 | 71 | 10 | - |
| Female/male | 6/3 | 57/14 | 7/3 | - |
| Age(year-old) | 69.6±8.1 | 68.8±8.1 | 72.1±8.7 | .654 |
| Pre-VAS | 7.7±1.0 | 7.4±0.7 | 7.4±0.5 | .840 |
| Post-VAS | 2.2±0.4 | 2.5±0.6 | 2.6±0.7 | .352 |

**Table 2. Demographics data and pre-operative spinal and pelvic sagittal parameters of healthy volunteers and OVCF patients.**

| | Group | | | | | P value | | | |
|---|---|---|---|---|---|---|---|---|---|
| | Healthy volunteers | OVCF patients | MT group | TL group | LU group | Control vs overall | Control vs MT | Control vs TL | Control vs LU |
| Number of patients | 26 | 90 | 9 | 71 | 10 | - | - | - | - |
| Female/male | 13/13 | 70/20 | 6/3 | 57/14 | 7/3 | - | - | - | - |
| Age(year-old) | 67.9±6.2 | 69.3±8.1 | 69.6±8.1 | 68.8±8.1 | 72.1±8.7 | .425 | 0.525 | .590 | .112 |
| TK | 22.6±9.1 | 37.8±15.8 | 40.7±9.6 | 38.7±16.9 | 28.4±7.8 | .000 | .000 | .000 | .086 |
| TLK | 4.9(0.9,11.4) | 27.3(16.4,40) | 22.9±9.2 | 33.3±18.4 | 12.5±11.1 | .000* | .000* | .000* | .177* |
| LL | -48.1±9.0 | -48.4(-56.3,-37.4) | -57.4±5.7 | -46.5±14.9 | -47.7±13.4 | 0.934* | .007 | .599 | .917 |
| PT | 17.2±7.0 | 24.0±8.7 | 16.7±6.4 | 24.8±8.9 | 24.6±6.6 | .000 | .850 | .000 | .007 |
| SS | 34.3±6.4 | 30.0±8.9 | 35.0±7.0 | 28.9±9.0 | 33.7±7.0 | .023 | .785 | .006 | .804 |
| PI | 51.6±6.4 | 52.2(47.7,61.3) | 52.9±7.6 | 52.0 (47.1,61.2) | 58.3±11.6 | 0.408* | .623 | .552* | .112 |
| PI minus LL | 3.5±8.7 | 9.3±14.2 | -4.5±8.2 | 7.1±13.6 | 10.6±7.9 | .014 | .022 | .213 | .032 |
| SVA | -2.0±18.5 | 14.5(-4.8,38.2) | 11.5±21.7 | 20.7±36.9 | 18.9±44.5 | .003* | .082 | .000 | .182 |
| SSA | 126.3±6.3 | 119.2±9.6 | 124.3±6.5 | 118.0±9.6 | 123.8±10.5 | .001 | .435 | .000 | .399 |
| TPA | 11.7±5.8 | 18.7(12.8,22.6) | 11.1 (9.0,14.3) | 19.6±8.4 | 19.6±5.1 | .000* | 0.985* | .000 | .001 |

*p-value derived using Mann-Whitney U test for nonparametric unpaired data

volunteers. Compared with that of the controls, the TK, TLK and LL of the MT group were significantly larger. There was no significant difference in pelvic parameters or global sagittal parameters between the MT group and the volunteers. Significant differences were observed between the LU group and the volunteers in terms of the PT and TPA. The other parameters had no distinctive difference. Demographic characteristics and sagittal parameters of the controls and all patients prior to the operation are reported in Table 2.

## Postoperative radiological measures

The VAS score of all patients decreased from 7.4±0.7 preoperatively to 2.5±0.6 postoperatively, and the pain was significantly relieved (p < 0.01). The results of the comparison between the preoperative and postoperative conditions are shown in Table 3. In the total patient group, significant increases in LL, SS and SSA were observed (p < 0.05). And there was also a significant decrease in PT, PI minus LL, TLK, local kyphosis Cobb angle, and TPA (p < 0.05).

In the TL group, the PT decreased from 24.8±8.9 to 22.7±7.4 and the SS increased from 28.9±9.0 to 30.6±8.5 after the operation (p < 0.01). The TLK and local kyphosis Cobb angle decreased from 33.3±18.4 and 23.3±15.6 to 29.3±16.3 and 18.3±13.6, respectively. The LL increased from -46.5±14.9 to -49.5±13.4, and there were significant differences in all local sagittal parameters except for the TK. Among the global sagittal parameters, the SSA increased from 118.0±9.6 to 120.5±9.5 and the TPA decreased from 19.6±8.4 to 17.1±7.2 after PKP (p < 0.001). Although the SVA decreased from 20.7±36.9 to 13.4±43.9 postoperatively, the difference was not significant (p = 0.094).

In the MT group, only TLK decreased from 22.9±9.2 to 19±9.6 after PKP. Other parameters were not statistically different after surgery.

In the LU group, the local kyphosis Cobb angle (p = 0.047) and SSA (p = 0.043) increased significantly after surgery. The PI minus LL was significantly reduced after surgery (p = 0.04).

**Table 3. Comparison of spinal and pelvic parameters of patients with OVCF between pre- and post-treatment.**

| Region | Parameters | Overall patient(90) | | | MT group(9) | | | TL group(71) | | | LU group (10) | | |
|---|---|---|---|---|---|---|---|---|---|---|---|---|---|
| | | Pre-operative | Post-operative | P value | Pre-operative | Post-operative | P value | Pre-operative | Post-operative | P value | Pre-operative | Post-operative | P value |
| Thoracolumbar | local kyphosis Cobb angle | 24±16.1 | 17.7 (9.3,28.5) | .000* | 25.1±14.2 | 22.0±12.95 | .094 | 23.3±15.6 | 18.3±13.6 | .000 | 27.5±21.8 | 30.96±21.6 | .047 |
| | TK | 37.8±15.8 | 36.7±14.3 | .066 | 40.7±9.6 | 36.5±11.2 | .068 | 38.7±16.9 | 37.9±15.1 | .207 | 28.4±7.8 | 28.2±5.9 | .892 |
| | TLK | 27.3 (16.4,40.0) | 23.4 (16.2,35.2) | .000* | 22.9±9.2 | 19.0±9.6 | .040 | 33.3±18.4 | 29.3±16.3 | .000 | 12.5±11.1 | 11.9±10.8 | .631 |
| | LL | -48.4 (-56.3,-37.4) | -50.2±12.9 | .001* | -57.4±5.7 | -55.8±6.5 | .255 | -46.5 ±14.9 | -49.5 ±13.4 | .000 | -47.7 ±13.4 | -49.99 ±13.2 | .079 |
| Pelvic | PT | 23.97±8.7 | 22.1±7.7 | .001 | 16.7±6.4 | 15.5±7.7 | .396 | 24.8±8.9 | 22.7±7.4 | .001 | 24.6±6.6 | 23.9±7.8 | .484 |
| | SS | 30.0±8.9 | 31.7±8.5 | .003 | 35.0±7.0 | 37.2±6.0 | .170 | 28.9±9.0 | 30.6±8.5 | .008 | 33.7±7.0 | 34.2±8.4 | .642 |
| | PI | 52.2 (47.7,61.3) | 53.9±10.3 | .106* | 52.9±7.6 | 52.7±7.8 | .451 | 52.0 (47.1,61.2) | 53.4±10.3 | .171* | 58.3±11.6 | 58.1±11.6 | .589 |
| | PI-LL | 6.29±13.1 | 3.7±10.9 | .000 | -4.5±8.2 | -3.1±10.1 | .281 | 7.1±13.6 | 3.9±11.1 | .000 | 10.6±7.9 | 8.2±8.3 | .040 |
| Global | SVA | 14.5 (-4.8,38.2) | 5.8 (-17.6,38.8) | .128* | 11.5±21.7 | 17.2±38.8 | .493 | 20.7±36.9 | 13.4±43.9 | .094 | 18.9±44.5 | -0.5 (-17.6,26.2) | .285* |
| | SSA | 119.3±9.6 | 121.7±9.5 | .000 | 124.3±6.5 | 129.2 (123.9,129.8) | .173* | 118.0±9.6 | 120.5±9.5 | .000 | 123.8 ±10.5 | 126.2±9.7 | .043 |
| | TPA | 18.7 (12.8,22.6) | 16.7±7.1 | .000* | 11.1 (9.0,14.3) | 11.8±6.0 | .483* | 19.6±8.4 | 17.1±7.2 | .000 | 19.6±5.19 | 18.2±5.8 | .080 |

*p-value derived using Wilcoxon signed rank test for nonparametric paired data

The correlations between the spinal and pelvic parameters in OVCFs preoperatively are shown in Table 4.There was a moderate correlation among the SVA, SSA and TPA (p<0.01). The TPA was positively correlated with the PT (r = 0.862) and PI-LL (r = 0.672, and the SSA was positively correlated with the SS (r = 0.801) and negatively correlated with the LL (r = -0.672) (p<0.01). Based on these results, the SSA is affected by the parameters of the pelvis and spine. The TPA is mainly affected by the pelvic parameters. There was no significant correlation between the SVA and most of the spinal and pelvic parameters.

**Table 4. The correlation between the preoperative spinal and pelvic parameters in OVCFs.**

| | SSA | TPA | Cobb | TK | TLK | LL | PT | SS | PI | PI-LL |
|---|---|---|---|---|---|---|---|---|---|---|
| SVA | -0.482* | 0.471* | -0.057 | -0.056 | 0.086 | 0.388* | 0.006 | 0.021 | 0.012 | 0.434* |
| SSA | | -0.411* | -0.142 | -0.183 | -0.548* | -0.672* | -0.285* | 0.801* | 0.484* | -0.359* |
| TPA | | | 0.171 | 0.116 | 0.221 | 0.229 | 0.862* | -0210 | 0.542* | 0.672* |
| Cobb | | | | 0.441* | 0.479* | -0.147 | 0.315* | -0.198 | 0.067 | -0.110 |
| TK | | | | | 0.678* | -0.542* | 0.227 | -0.215 | 0 | -0.594* |
| TLK | | | | | | 0.052 | 0.283* | -0.550* | -0.257 | -0.143 |
| LL | | | | | | | 0.075 | -0.592* | -0.471* | 0.728* |
| PT | | | | | | | | -0.334* | 0.543* | 0.503* |
| SS | | | | | | | | | 0.599* | -0.182 |
| PI | | | | | | | | | | 0.261 |

*Correlation is significant at the 0.01 level (2-tailed)

## Power analysis

With effect size of 0.8 and 0.05 level of statistical significance, the TL group (n = 71) achieved a power of 0.99. Thus, the TL group was sufficiently powered to detect the effect of PKP on sagittal balance in patients with OVCF. However, power in the non-TL groups ranged from 0.55 to 0.65. Based on these findings, there is insufficient data to investigate the parameters in the non-TL groups. Thus, additional studies with proper large-scale cohorts are still warranted.

## Discussion

Global sagittal balance is an optimal state of equilibrium, during which the standing position is maintained with a horizontally balanced posture, minimal energy expenditure, and minimal ligament discomfort [4, 28]. The most important aspects of sagittal balance are to achieve harmony between the sagittal parameters of the spine and the pelvis and to maintain the axis of gravity at its natural location with minimal energy consumption [28–32].

When the local sagittal alignment of the spine is abnormal, the body will initiate multiple compensatory mechanisms to maintain global balance. Compensatory mechanisms have been found in the pelvis, spine and lower extremities[33]. When a spinal disease affects the spinal compensations, the main manifestations of the compensatory process are a pelvic posterior rotation and knee flexion compensation [34].

When the deformity gradually worsens beyond the compensatory capacity of the pelvis, spine and lower limbs and the body cannot maintain balance by increasing muscle strength, there will be a failure to achieve a horizontal gaze and to maintain alignment of the gravity line, resulting in a sagittal imbalance.

In clinical practice, many metric and angular parameters of the full-length lateral radiograph of the spine have been used to assess the sagittal balance. Global sagittal balance is typically determined by measuring the SVA [4, 35].

With the increasing use of the SVA in clinical research, a shortcoming has gradually emerged. The limitation of the SVA is that it is influenced by the patient's position and pelvic rotation [36]. For instance, when a large thoracolumbar kyphosis occurs, the spine could maintain balance through muscle adjustments and the SVA could achieve normal values, but the patient's pain symptoms would be more obvious. Therefore, the SVA does not truly reflect the structure of the spine and the severity of the patient's symptoms. In addition, the SVA is a linear parameter that must be calibrated proportionally due to the influence of the X-ray projection distance, and the deviation is relatively large.

To avoid these drawbacks, we and other researchers propose to use angular parameters such as the SSA and TPA[20, 33].

Roussouly et al [20] proposed the concept of the SSA. In a normal population, the mean value of this angle is 135 ± 8 [20, 33]. The SSA has been used not only to assess the global sagittal alignment of the spine above the pelvis but also to reflect the size of the entire kyphosis. The SSA value decreases when kyphosis is present in the spine.

Protopsaltis et al [27] proposed a new parameter reflecting sagittal balance: the TPA. The TPA integrates global and local spino-pelvic sagittal balance information and reflects the compensatory mechanism of the spine and pelvis. Similar to the SSA, the TPA is also an angular parameter. These parameters do not need to be calibrated proportionally on imaging data, and their errors are smaller than those of metric parameters such as the SVA.

Our study found that the SSA and TPA displayed greater correlations with the sagittal parameters of the spine and pelvis than the SVA. Our results showed that the TPA was positively correlated with the PT and PI-LL, and the SSA was positively correlated with the SS and negatively correlated with the LL. These results suggest that the SSA and TPA are affected by

the parameters of the pelvis and spine. There was no significant correlation between the SVA and most of the spino-pelvic sagittal parameters. Therefore, the clinical reference values of these two parameters are better than that of the SVA.

A spinal sagittal imbalance can be caused by many spinal diseases, such as spinal deformity and spinal degeneration. In recent years, some scholars have begun to pay attention to the sagittal imbalances caused by osteoporotic vertebral compression fractures.

Sutipornpalangkul et al[37] confirmed that patients with OVCFs had anterior wedge deformities, leading to the progression of kyphotic deformity and an anterior shift in the center of gravity, and ultimately causing a spinal sagittal imbalance.

Le Huec JC et al [38] reported that patients with OVCFs showed a worse global sagittal alignment and decreased quality of life. The number and severity of vertebral compression fractures had a negative influence on global sagittal balance.

In our study, we confirmed that after fracture, especially thoracolumbar fracture, the kyphosis deformity worsened, the C7 plumb line shifted forward, and the sagittal balance was mainly compensated by pelvic retroversion. Some patients with severe fractures could not be corrected by compensation, resulting in a sagittal imbalance. These patients showed more obvious symptoms than simple low back pain caused by vertebral fractures. The most common symptoms were the tendency to tilt forward when standing or walking and the failure to walk on his/her own without support from the front of the body.

Kyphoplasty is a minimally invasive treatment for OVCFs. In this case, whether the classic surgical method of OVCF, PKP, can be used to restore the global sagittal balance is still controversial [39–43].

Some scholars have confirmed that PKP not only alleviates the pain caused by a fracture but also improves the sagittal balance by restoring the anterior height of the vertebral body and improving the local kyphosis deformity [8, 37].

However, few detailed studies have investigated the effects of kyphoplasty on total spinal alignment or global sagittal balance [8].

Kanayama et al[39] analyzed 56 patients with OVCF who underwent PKP. After 32 months of follow-up, the research group found that PKP contributed to immediate pain relief but did not improve the global sagittal alignment after OVCF. The researchers concluded that PKP should be solely used to address pain or the nonunion of an OVCF and could not be expected to restore the global sagittal alignment.

Sutipornpalangkul et al[37] analyzed 17 patients with OVCF who underwent PKP and concluded that, kyphoplasty did not play a role in improving the overall alignment of the spine for the treatment of OVCF. However, kyphoplasty did demonstrate regional improvement of the OVCF. The researchers supposed that a multiple-level kyphoplasty might improve overall sagittal balance. The main reason may be the cumulative improvements in the degrees of correction of the kyphotic angle.

However, Yokoyama et al [8] analyzed 21 patients with OVCF treated with PKP and showed that PKP not only alleviated the pain associated with fractures but also significantly improved sagittal imbalance.

In our study, we found that patients with OVCFs had a reduction in kyphosis after PKP and that the pelvic posterior retroversion was significantly restored after surgery. Some patients with sagittal imbalance regained sagittal balance or at least achieved a compensatory balance. According to the analysis of our data, PKP could improve the sagittal balance parameters, including the overall balance parameters. Osteoporotic compression fractures of the thoracolumbar segment achieved the best improvement among the three fracture groups.

We found that although fractures of the thoracolumbar segment had the largest incidence and were associated with the most severe kyphosis, the improvement of the thoracolumbar

segment after PKP was the greatest. This improvement may be mainly attributed to the anatomical and biomechanical characteristics of the thoracolumbar segment. The thoracolumbar spine section generally extends from T10 to L2, which includes the junction of the thoracic and lumbar segments. This section carries a large spinal load and is extremely susceptible to damage, and a certain degree of kyphosis deformity occurs upon injury.

The compressive stress that occurs during an injury is likely to cause the collapse of an anterior vertebral fracture. The superior endplate is often involved in an osteoporotic vertebral fracture due to the unique structure and different distribution of trabecular bone across the vertebral body [7, 44, 45]. After the PKP treatment, the stability of the anterior and middle columns of the thoracolumbar segments improved significantly, and the anterior support function recovered partially. The reverse injury mechanism gradually corrected a part of the sagittal imbalance. Another potential explanation for the improvement in sagittal imbalance after thoracolumbar fracture is the large amount of sagittal loss after the thoracolumbar fracture.

Osteoporotic vertebral compression fractures mainly occurred in the thoracolumbar region (T10-L2) in several previous studies[46–49]. Liu T et al studied 77 patients with single-segment OVCF, of which 77.9% occurred in the thoracolumbar region[50]. Kong LD et al studied 53 patients with OVCF and 75.5% of fractures occurred in the thoracolumbar region [51]. In our study, this probability was 78.9%. Therefore, the number of patients in non-TL group was relatively small. And some of these patients also had fractures in the thoracolumbar region at the same time. In order to avoid confounding factors in the statistical analysis, we excluded patients with fractures in two or more different regions. Therefore, the number of patients in non-TL group was even smaller, which may have resulted in a higher statistical bias. Thus, additional studies with high-quality and large-scale are still warranted.

## Conclusion

PKP is an effective treatment for osteoporotic thoracolumbar vertebral compression fractures. When OVCFs occurred in the thoracolumbar region (T10-L2), PKP can not only relieves the low back pain caused by fractures but also corrects the pelvic posterior rotation that occurs during sagittal compensatory balance within 2–3 days. PKP can significantly improves the angular parameters (TPA and SSA) caused by vertebral fractures and improves the overall sagittal alignment.

Osteoporotic vertebral compression fractures mainly occur in the thoracolumbar region, affecting the spino-pelvic alignment and global sagittal balance to a greater extent than in the MT region and LU region. On the other hand, among the three groups, the improvement of sagittal balance parameters was greatest in patients with a fracture in the thoracolumbar region.

### Drawbacks of this study

1. Due to the incidence rate, the numbers of fracture cases in the upper thoracic vertebrae and lower lumbar vertebrae were small, which may have caused statistical errors and lack of persuasiveness.

2. Because most of the patients were elderly, their activity was limited, and their self-care ability was poor. When the whole spine was taken into account, the most satisfactory standing posture was not always able to be achieved, resulting in some errors.

3. All patients were from the same surgical group, and the surgical procedures were basically the same, but there were inevitable surgical differences among the three surgeons.

## Supporting information

**S1 Table. Demographics data and spino-pelvic sagittal parameters of OVCF patients.**
(XLSX)

**S2 Table. Demographics data and spino-pelvic sagittal parameters of healthy volunteers.**
(XLSX)

**S1 Text. The STROBE statement—Checklist.**
(DOCX)

## Acknowledgments

We would like to thank Fanxiao Liu for his assistance with the professional linguistic editing. Further appreciation was expressed for Haoran Qi's assistance in data collection.

## Author Contributions

**Conceptualization:** Zhong Cao, Jianmin Sun.

**Data curation:** Zhong Cao, Guodong Wang, Zhiyong Liu.

**Formal analysis:** Zhong Cao, Guodong Wang.

**Investigation:** Zhong Cao, Guodong Wang, Wenpeng Hui.

**Methodology:** Zhong Cao, Guodong Wang, Wenpeng Hui, Bo Liu.

**Software:** Zhong Cao, Bo Liu.

**Supervision:** Jianmin Sun.

**Validation:** Zhong Cao, Jianmin Sun.

**Visualization:** Zhong Cao, Wenpeng Hui, Jianmin Sun.

**Writing – original draft:** Zhong Cao, Guodong Wang, Jianmin Sun.

**Writing – review & editing:** Zhong Cao, Jianmin Sun.

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
