## [Decision Letter · Decision Letter 0]

20 Sep 2019

PONE-D-19-23099

The influences of the site of osteoporotic vertebral compression fracture and percutaneous kyphoplasty treatment on spino-pelvic alignment and global sagittal balance

PLOS ONE

Dear Mr. Sun,

Thank you for submitting your manuscript to PLOS ONE. After careful consideration, we feel that it has merit but does not fully meet PLOS ONE’s publication criteria as it currently stands. Therefore, we invite you to submit a revised version of the manuscript that addresses the points raised during the review process.

The study was well received by the reviewers. Some salient points that came up in this initial review include concerns about patient selection in the study cohort that may produce some bias; the lordosis measurements need to be confirmed as well, and finally, the reviewer panel also urges to look carefully into the statistics to draw the correct conclusions.

We would appreciate receiving your revised manuscript by Nov 04 2019 11:59PM. To enhance the reproducibility of your results, we recommend that if applicable you deposit your laboratory protocols in protocols.io, where a protocol can be assigned its own identifier (DOI) such that it can be cited independently in the future. For instructions see: http://journals.plos.org/plosone/s/submission-guidelines#loc-laboratory-protocols

We look forward to receiving your revised manuscript.

Kind regards,

Alejandro A. Espinoza Orías, PhD

Academic Editor

PLOS ONE

Reviewers' comments:

Reviewer's Responses to Questions

**Comments to the Author**

1. Is the manuscript technically sound, and do the data support the conclusions?

Reviewer #1: Yes

Reviewer #2: Yes

Reviewer #3: No

2. Has the statistical analysis been performed appropriately and rigorously? 

Reviewer #1: I Don't Know

Reviewer #2: Yes

Reviewer #3: Yes

3. Have the authors made all data underlying the findings in their manuscript fully available?

Reviewer #1: Yes

Reviewer #2: Yes

Reviewer #3: No

4. Is the manuscript presented in an intelligible fashion and written in standard English?

Reviewer #1: Yes

Reviewer #2: Yes

Reviewer #3: Yes

5. Review Comments to the Author

Reviewer #1: My review for PLOS One article:

The influences of the site of osteoporotic vertebral compression fracture and percutaneous kyphoplasty treatment on spino-pelvic alignment and global sagittal balance

Mon Sep 16

20,000 characters max

Some although not all correctable English language errors are included below.

Full Title: Consider re-stating as a hypothesis, something like: Percutaneous Kyphoplasty for Osteoporotic Vertrebral Compression Fractures Improves Spino-Pelvic and Sagittal Alignment Maximally in the Thoracolumbar Region.

For the 3 groups of vertebral compression fracture regions, MT, TL, and LU, please mention the words represented by these abbreviations (or rename them to something more intuitive?)

The authors refer to ‘PKP’ as a surgery, whereas I would use the word ‘procedure’ to describe it. This may be confirmed with subspecialists performing the procedure, interventional radiologists in my experience.

‘Eldery patients’: Can you include the age range?

Additional specific suggested edits/ideas listed below:

Line #

25 The sentence including ‘it’s still blank on’ is unclear

29 Change ‘OCVF underwent’ to ‘OCVF who underwent’

31 Please explain here meaning of the 3 group abbreviations MT, TL, LU

33 Radiographs, not radiographies

44 ‘When performed’ not ‘while available

72 ‘with’ Shandong University

75 Mention why only compression fractures of less than 80% an inclusion criteria.

80 begin this line with ‘and’

127 use ‘before and after’ instead of ‘from before to after’

159 ‘increased significantly’ associated with what? Please specify. This is in the pre-operative section so I assume not due to treatment.

305 Using ‘from T10 – L2’ is convention in USA or write it out fully in English.

310 Briefly clarify where the ‘rib support is lost’—I assume from T12 to L1, or lumber compared to thoracic regions.

325 How do we know that ‘support strengthened’ after PKP? It makes sense that the additional improvements in pain and strength improving can be objectively measured, but unsure what is meant by ‘support strengthened’.

Reviewer #2: The authors of the study pose a relevant question that has not been answered sufficiently in the literature. The study is generally well performed with appropriate methods. The paper titled “The influences of the site of osteoporotic vertebral compression fracture and percutaneous kyphoplasty treatment on spino-pelvic alignment and global sagittal

balance” is generally well written, however, there are some minor issues to address:

-Please provide list of abbrevations

Introduction:

-Line 43-45 (Conclusion): Language issue – please revise

Materials + Methods:

-Line 79: .. the whole spine with pelvis and femoral heads…

Results:

-Line 157: please provide possible explanation why LL is not different in healthy volunteer group vs. LU group. In conclusions, you describe “…improve the local kyphosis caused by fractures…”. Your data of lumbar lordosis does not support that assumption. Please provide additional explanation or change conclusion.

-Please provide a numerical definition of what you consider as strong vs. moderate vs. weak statistical correlation

-Table 4: I am not sure what you mean by significant correlation: e.g. a correlation of SSA vs. PT of -0.285 does not seem significant to me. please clarify.

Discussion:

Line 209: “The experience…” Language – please revise

Line 238: we and others propose

Line 251-253: a correlation of TPA and PI of 0.542 will usually be considered as weak. Same applies for SSA and PI when r=0.484 and LL and TLK.

Line 290: suppose – Language- please revise

Line 308-319: this does not seem relevant to the study. consider removal.

Line 320-322: this does not seem relevant to the study. consider removal.

Line 327-329: unclear language – please revise

Conclusion

Line 335: “PKP can significantly improve the angular parameters (TPA, SSA) caused by vertebral fractures and improve the overall sagittal alignment.” Did you find any correlation of improved TPA and/or SSA to improved VAS?

Reviewer #3: First of all congratulations to the authors for this excellent work. I think that you have worked hard and the ammount of data that you have is so good.

Prior to the publication, I would need to know some details of the work:

- kyphoplasty has shown to reduce the pain related with the osteoporotic vertebral fracture faster than the conservative treatment, but it has not shown clearly to reduce the disbalance. Your work is based on "the influence of the site of osteoporotic fracture AND percutaneous kyphoplasty", and you finish your discussion with the phrase "PKP is an effective treatment for osteoporotic thoracolumbar vertebral compression fractures". And then you follow: "... also improve the local kyphosis caused by fractures and correct the pelvic posterior rotation that occurs during sagittal compensatory balance". In my opinion, you don't have evidence enough to say that, for this reason: you have not compared the balance of people with a vertebral fracture treated with conservative management vs PKP treatment. You can define what deformity do you expect after a fracture treated with PKP but you can't say that improves the kyphosis.

It's clear that the kyphoplasty improves the heigh of the vertebral body, but It has been related with adjacent vertebra fractures, wich could produce kyphosis too. That's why you must compare the kyphosis after the conservative treatment to say that PKP improves the kyphosis, or, at least, to define what do you mean with "local kyphosis", and mentioning that it can produce an hiper-kyphosis in other segments.

- When were the post-op X-ray taken? That's probably the most important factor of the study. PKP has shown to heel the fractures and to reduce the pain faster than conservative treatment, and obviously, when the patient has pain, he tries to compensate the spine balance, what could produce an hiper-kyphosis that would disappear after the consertative management when the pain is not so strong. You must compare the effect of the PKP in the spine balance in a chronic timeline, cause the spine balance is important to evaluate the chronic back pain, not the acute back pain. In other words, when we have neck pain after a car accident, we can modify the cervical lordosis, but it's not a surgical indication to correct it, cause when the pain will dissappear, the lordosis will come again to its normal position.

- Why did you select just patients on wich the injured vertebrae compression was less than 80%? How could you explain the ammount of kyphosis produced before surgery only with a maxium of 20% compression? Could the pain be an explanation for that?

- How did you define the parametric statistics for the comparisons? On which values the normality test was applied? It's difficult to get a Normal distribution in the MT and LU group just with 9 and 10 patients respectively.

- You must be careful when you say "there was no significant difference in... when p value is higher than 0,05. As you probably know, when the p value is higher than the a value, you cannot establish that there are no differences: you can say that in your data you don't have found them, but with the n value of the MT and the LU group, you could be producing a Type II error.

- Finally, this is a retrospective study on which you assume as exclusion criteria patients with "hip and knee joint limitations". How did you verify that the population of your study didn't have symptoms or limitations due to hip or knee osteoarthritis

Finally, I would like to verify the statistics and data personally.

Thanks for all these clarifications.

6. PLOS authors have the option to publish the peer review history of their article (what does this mean?). If published, this will include your full peer review and any attached files.

Reviewer #1: No

Reviewer #2: No

Reviewer #3: No

---

## [Author Response · Author response to Decision Letter 0]

29 Oct 2019

Reviewer's Responses to Questions：

1. Is the manuscript technically sound, and do the data support the conclusions?

Reviewer #1: Yes

Reviewer #2: Yes

Reviewer #3: No

Response: We appreciate the reviewer’s comments. We asked professional statisticians to conduct a statistical analysis of our data again, and made corresponding changes to our conclusions.

2. Has the statistical analysis been performed appropriately and rigorously? 

Reviewer #1: I Don't Know

Reviewer #2: Yes

Reviewer #3: Yes

Response: Thanks to the reviewers for their affirmation of our statistical methods. All of our data statistics programs are consulted by statistical professionals, and it is believed that the statistical methods applied should be appropriate and rigorous.

3. Have the authors made all data underlying the findings in their manuscript fully available?

Reviewer #1: Yes

Reviewer #2: Yes

Reviewer #3: No

Response: Thanks for the two reviewers. All the data involved in our manuscript has been uploaded as an attachment during the submission. Please see the attached S1 Table and S2 Table.

4. Is the manuscript presented in an intelligible fashion and written in standard English?

Reviewer #1: Yes

Reviewer #2: Yes

Reviewer #3: Yes

Response: Thanks for the reviewer's recognition of our manuscript language.

 5. Review Comments to the Author

Responses to Reviewer #1 

We would like to extend a special thank you for your helpful comments and constructive criticism.

1) Comment：Full Title: Consider re-stating as a hypothesis, something like: Percutaneous Kyphoplasty for Osteoporotic Vertebral Compression Fractures Improves Spino-Pelvic and Sagittal Alignment Maximally in the Thoracolumbar Region.

 Response: Thanks for your valuable advice. We have made the correction according to the Reviewer’s comments. 

2) Comment：For the 3 groups of vertebral compression fracture regions, MT, TL, and LU, please mention the words represented by these abbreviations (or rename them to something more intuitive?)

Response: We are very sorry to have missed the full name of MT, TL, and LU. In our manuscript, MT, TL, and LU are short for Main Thoracic, Thoracolumbar, and Lumbar, respectively. According to the reviewer's suggestion, we added the full name and the corresponding abbreviation of three groups in the manuscript.

3) Comment：The authors refer to ‘PKP’ as a surgery, whereas I would use the word ‘procedure’ to describe it. This may be confirmed with subspecialists performing the procedure, interventional radiologists in my experience.

Response: Thanks for the kind advices. In our hospital, PKP is performed by spinal surgeons in the operating room, so we are used to calling it "surgery." Considering the Reviewer’s warm suggestion, we agree to use ' procedure ' to describe it in the manuscript. 

4) Comment：‘Elderly patients’: Can you include the age range? 

Response: We appreciate the reviewer’s comments. In the manuscript, we were too vague about ‘Elderly patients’. In our study, the control group was no younger than 59 years old and we have modified it accordingly.

 5) Comment：Additional specific suggested edits/ideas listed below:

Line #

25 The sentence including ‘it’s still blank on’ is unclear

29 Change ‘OCVF underwent’ to ‘OCVF who underwent’

31 Please explain here meaning of the 3 group abbreviations MT, TL, LU

33 Radiographs, not radiographies

44 ‘When performed’ not ‘while available

80 begin this line with ‘and’

127 use ‘before and after’ instead of ‘from before to after’

305 Using ‘from T10 – L2’ is convention in USA or write it out fully in English.

Response: We are very sorry for the mistakes. In response to the above language expressions, we have made corresponding modifications according to the comments of reviewers. Thank you! 

6) Comment：Line 72 ‘with’ Shandong University

Response: Thanks for your valuable advice. As Shandong provincial hospital affiliated to Shandong University is the official name of our hospital, so it's the best way to use the official name to represent my institution. Many thanks!

7) Comment：Line 75 Mention why only compression fractures of less than 80% an inclusion criteria.

Response: Thanks for your reminding. The vertebral compression ratio of less than 80% is one of the surgical indications that most orthopedic surgeons in our country have for patients with osteoporotic vertebral compression fractures [1]. We believe that if the compression ratio of the vertebral body is greater than 80%, the difficulty of PKP puncture is significantly increased, and cement leakage in percutaneous kyphoplasty is more likely to occur. 

[1] Yin P, Ma YZ, Ma X, Chen BH, Hong Y, Liu BG, et al. Guidelines for the treatment of osteoporosis vertebral compression fractures[ J].Chinese Journal of Osteoporosis, 2015, 21(06): 643-648.

8) Comment：Line 159 ‘increased significantly’ associated with what? Please specify. This is in the pre-operative section so I assume not due to treatment.

Response: We are very sorry for the mistakes. Our original intention was that the TK, TLK, and LL were significantly larger in the MT group compared with the control group, and the expression in the manuscript was inaccurate. We have made correction according to the Reviewer’s comments. Thank you!

9) Comment：Line 310 Briefly clarify where the ‘rib support is lost’—I assume from T12 to L1, or lumber compared to thoracic regions.

Response: Thanks for your valuable advice. We believe that from the thoracic spine to the lumbar spine, the fixation of the ribs gradually diminishes until it disappears. Because another reviewer believes that lines 308-322 do not seem relevant to the study, so we corrected the description.

10) Comment：Line 325 How do we know that ‘support strengthened’ after PKP? It makes sense that the additional improvements in pain and strength improving can be objectively measured, but unsure what is meant by ‘support strengthened’.

Response: Special thanks to you for your good comments. ‘support strengthened’ after PKP has no literature support yet, we have corrected the description.

Responses to Reviewer #2

We would like to extend a special thank you for your helpful comments and constructive criticism.

1) Comment：Please provide list of abbreviations.

Response: Thanks for your valuable advice. We have added a list of abbreviations and noted them in the manuscript.

2) Comment：Introduction: Line 43-45 (Conclusion): Language issue – please revise

Materials + Methods: Line 79: …the whole spine with pelvis and femoral heads…

Response: Considering the Reviewer’s suggestion, we have made corresponding modifications. Please review again, Thank you! 

3) Comment：Results: Line 157: please provide possible explanation why LL is not different in healthy volunteer group vs. LU group. In conclusions, you describe “…improve the local kyphosis caused by fractures…”. Your data of lumbar lordosis does not support that assumption. Please provide additional explanation or change conclusion.

Response: Thanks for your valuable advice. Our statistics do show no significant difference in LL between LU group and control group. We re-read the full-length X-ray of the spine of all patients in the LU group. We found that all 10 patients in the LU group were just mild compression fractures or biconcave compression fractures. There was no obvious wedge deformation in the anterior and posterior columns of the injured vertebral body, so the LL did not change much. It may explain why there is no difference in LL between the LU group and control group. Of course, this situation is mainly related to the low incidence of OVCF in the lower lumbar vertebrae in clinical practice, and thus the number of patients enrolled in the group is relatively small. Inevitably there will be some statistical errors, causing data bias. 

The conclusion of "... improve the local kyphosis by products ..." cannot be directly derived from this data. We analyzed the data and found that PKP did not improve the kyphosis of the MT and LU groups, but significantly improved the kyphosis of the TL group. These data can be obtained from the comparison of TLK, LL, and local kyphosis Cobb angle data before and after PKP in TL group. Therefore, we have made relevant amendments to the conclusions in order to be more objective and fair. Thank you for your reminder.

4) Comment：Please provide a numerical definition of what you consider as strong vs. moderate vs. weak statistical correlation

-Table 4: I am not sure what you mean by significant correlation: e.g. a correlation of SSA vs. PT of -0.285 does not seem significant to me. please clarify.

Line 251-253: a correlation of TPA and PI of 0.542 will usually be considered as weak. Same applies for SSA and PI when r=0.484 and LL and TLK.

Response: We appreciate the reviewer’s comments. The strength of the correlation was determined using the following guide for the absolute value: 0.00–0.19 “very weak,” 0.20–0.39 “weak,” 0.40–0.59 “moderate,” 0.60–0.79 “strong,” ,0.8-1.0 “very strong,”. We have re-written this part according to the Reviewer’s suggestion. Thanks a lot.

5) Comment：Discussion:

Line 209: “The experience…” Language – please revise

Line 238: we and others propose

Line 290: suppose – Language- please revise

Line 308-319: this does not seem relevant to the study. consider removal.

Line 320-322: this does not seem relevant to the study. consider removal.

Line 327-329: unclear language – please revise

Response: Thank you for your valuable comments. We have made the corresponding changes according to the Reviewer’s comments.

6) Comment：Conclusion

Line 335: “PKP can significantly improve the angular parameters (TPA, SSA) caused by vertebral fractures and improve the overall sagittal alignment.” Did you find any correlation of improved TPA and/or SSA to improved VAS? 

Response: Thank you for your great comments. We have also done relevant statistical analysis in the early stage. After statistical analysis, it was found that there was no strong statistical correlation between improved TPA or SSA and improved VAS. Therefore, this aspect was not mentioned in our manuscript. We supposed that VAS, as subjective data, was greatly influenced by the patients themselves, and the individual differences are relatively large, which may affect the accuracy of statistics.

Reviewer #3: 

We would like to extend a special thank you for your helpful comments and constructive criticism.

1) Comment：kyphoplasty has shown to reduce the pain related with the osteoporotic vertebral fracture faster than the conservative treatment, but it has not shown clearly to reduce the disbalance. Your work is based on "the influence of the site of osteoporotic fracture AND percutaneous kyphoplasty", and you finish your discussion with the phrase "PKP is an effective treatment for osteoporotic thoracolumbar vertebral compression fractures". And then you follow: "... also improve the local kyphosis caused by fractures and correct the pelvic posterior rotation that occurs during sagittal compensatory balance". In my opinion, you don't have evidence enough to say that, for this reason: you have not compared the balance of people with a vertebral fracture treated with conservative management vs PKP treatment. You can define what deformity do you expect after a fracture treated with PKP but you can't say that improves the kyphosis.

It's clear that the kyphoplasty improves the height of the vertebral body, but It has been related with adjacent vertebra fractures, which could produce kyphosis too. That's why you must compare the kyphosis after the conservative treatment to say that PKP improves the kyphosis, or, at least, to define what do you mean with "local kyphosis", and mentioning that it can produce an hiper-kyphosis in other segments.

Response: Thanks for your reminding. We have re-written this part according to the Reviewer’s suggestion.

PKP has been reported to improve the imbalance caused by OVCF (YOKOYAMA K, et al. 2015), but it is still on controversy. In clinical practice, we encountered some patients with sagittal imbalance caused by OVCF. The sagittal balance was greatly improved within 2-3 days after PKP, these findings raise concern. We retrospectively studied that PKP could indeed correct mild to moderate sagittal imbalance early, especially in patients with fractures of the thoracolumbar region. It is the purpose to do the study.

The improvement in the sagittal imbalance involved in our study occurred within 2-3 days after PKP, and we all knew that the conservative treatment group did not change significantly within 2-3 days. Therefore, we did not choose a conservative treatment group as a comparison. Because no other interventions were taken during this period, we believed that PKP was helpful in improving the sagittal imbalance. We analyzed the data and found that PKP did not improve the kyphosis of the MT and LU groups, but significantly improved the kyphosis of the TL group. These data could be obtained from the comparison of TLK, LL, and local kyphosis Cobb angle data before and after PKP in TL group. Therefore, we have made relevant amendments to the conclusions in order to be more objective and fair. Long-term follow-up of these patients is still ongoing, and we will add a conservative treatment group for long-term follow-up. Thank you for your suggestion.

Of course, we must admit that PKP has no effect on the improvement of severe kyphosis or severe sagittal imbalance. These need to be corrected by osteotomy and are not suitable for PKP treatment. Therefore, we need to explain that the local kyphosis involved in the study is mostly mild to moderate thoracolumbar kyphosis.

2) Comment: When were the post-op X-ray taken? That's probably the most important factor of the study. PKP has shown to heel the fractures and to reduce the pain faster than conservative treatment, and obviously, when the patient has pain, he tries to compensate the spine balance, what could produce an hiper-kyphosis that would disappear after the conservative management when the pain is not so strong. You must compare the effect of the PKP in the spine balance in a chronic timeline, cause the spine balance is important to evaluate the chronic back pain, not the acute back pain. In other words, when we have neck pain after a car accident, we can modify the cervical lordosis, but it's not a surgical indication to correct it, cause when the pain will disappear, the lordosis will come again to its normal position.

Response: We appreciate the reviewer’s comments.Our postoperative radiographs were taken 2-3 days after PKP treatment. What we wanted to confirm was that PKP could significantly improve the sagittal imbalance earlier. It was an extra finding when we used PKP to treat OVCF to relieve pain and improve quality of life.

At the same time, we play correlation statistics on the improvement of VAS and the improvement of sagittal parameters such as SSA and TPA. It had been confirmed that there was no strong correlation between them. It also indicated that the improvement of the patient's sagittal imbalance was not caused solely by pain relief. We suspected that it might be related to factors such as back muscles. We were currently conducting a prospective study and the results will be reported later.

3) Comment: Why did you select just patients on which the injured vertebrae compression was less than 80%? How could you explain the amount of kyphosis produced before surgery only with a maximum of 20% compression? Could the pain be an explanation for that? 

Response: Thanks for your valuable advice. The reason why the selected cases in our study are less than 80% of the vertebral compression is that PKP is not recommended for OVCF patients with more than 80% injured vertebrae compression in our country. If the degree of compression of the injured vertebra is greater than 80%, cement leakage is more likely to occur, and intraoperative puncture is difficult.

The kyphosis is not only caused by the wedge deformation of the vertebral body, but also by other factors. We speculate that imbalance of muscle strength is one of the causes of kyphosis when the compression of the injured vertebra is less than 20%. Our hypothesis is that after fracture, the muscle activity in the back of the spine is abnormal, the muscle strength is decreased, the muscle strength in front of and behind the spine is unbalanced, and the kyphosis is aggravated. Severe cases further cause sagittal imbalance. Of course, pain after a vertebral fracture may also be an important cause. 

4) Comment: How did you define the parametric statistics for the comparisons? On which values the normality test was applied? It's difficult to get a Normal distribution in the MT and LU group just with 9 and 10 patients respectively. 

You must be careful when you say "there was no significant difference in... when p value is higher than 0,05. As you probably know, when the p value is higher than the a value, you cannot establish that there are no differences: you can say that in your data you don't have found them, but with the n value of the MT and the LU group, you could be producing a Type II error.

Response:

Prerequisites for using parameter statistics: 1. The data satisfies the normality; 2. The variances between the groups of data are equal (satisfying the homogeneity of the variance). We use the Shapiro-Wilk normality test, and when the p-value of the test result is greater than 0.05, the data is considered to satisfy the normality. 

The incidence of OVCF in the upper thoracic and lower lumbar was relatively low. In addition, we also needed the patients to meet the inclusion and exclusion criteria, as well as underwent preoperative and postoperative full-length radiographs. Therefore, the number of cases which satisfied the condition was indeed small, which inevitably caused a certain statistical error. This is also the limitations of our research. 

From the statistical analysis, we were fortunate to find that most of the parameters of the MT group and the LU group met a normal distribution. For data that did not conform to the normal distribution, we used nonparametric statistics.

5) Comment: Finally, this is a retrospective study on which you assume as exclusion criteria patients with "hip and knee joint limitations". How did you verify that the population of your study didn't have symptoms or limitations due to hip or knee osteoarthritis?

Response: Our treatment group patients were all inpatients, and their hospital records detailed whether the patients had a history of hip and knee disease. And the patient's range of motion of hip and knee can be seen in the physical examination of the medical record.

6) Comment: Finally, I would like to verify the statistics and data personally.

Response: We are very pleased that you can help us verify the statistics and data. The original data of our research has been uploaded to the attachment during the submission. Please review attachment S1 Table and S2 Table. Thank you!

---

## [Decision Letter · Decision Letter 1]

4 Dec 2019

PONE-D-19-23099R1

Percutaneous Kyphoplasty for Osteoporotic Vertebral Compression Fractures Improves Spino-Pelvic Alignment and Global Sagittal Balance Maximally in the Thoracolumbar Region

PLOS ONE

Dear Mr. Sun,

Thank you for submitting your manuscript to PLOS ONE. After careful consideration, we feel that it has merit but does not fully meet PLOS ONE’s publication criteria as it currently stands. Therefore, we invite you to submit a revised version of the manuscript that addresses the points raised during the review process.

The manuscript has seen important improvements since the first review, but on this occasion the reviewer panel found minor grammar and writing issues that need to be addressed, preferably with the aid of a native English speaker, so that the message is appropriately transmitted to the readership of the Journal. Please follow the reviewers' recommendations when making these corrections.

We would appreciate receiving your revised manuscript by Jan 18 2020 11:59PM. To enhance the reproducibility of your results, we recommend that if applicable you deposit your laboratory protocols in protocols.io, where a protocol can be assigned its own identifier (DOI) such that it can be cited independently in the future. For instructions see: http://journals.plos.org/plosone/s/submission-guidelines#loc-laboratory-protocols

We look forward to receiving your revised manuscript.

Kind regards,

Alejandro A. Espinoza Orías, PhD

Academic Editor

PLOS ONE

Reviewers' comments:

Reviewer's Responses to Questions

**Comments to the Author**

1. If the authors have adequately addressed your comments raised in a previous round of review and you feel that this manuscript is now acceptable for publication, you may indicate that here to bypass the “Comments to the Author” section, enter your conflict of interest statement in the “Confidential to Editor” section, and submit your "Accept" recommendation.

Reviewer #1: All comments have been addressed

Reviewer #4: (No Response)

2. Is the manuscript technically sound, and do the data support the conclusions?

Reviewer #1: Yes

Reviewer #4: Partly

3. Has the statistical analysis been performed appropriately and rigorously? 

Reviewer #1: I Don't Know

Reviewer #4: Yes

4. Have the authors made all data underlying the findings in their manuscript fully available?

Reviewer #1: Yes

Reviewer #4: Yes

5. Is the manuscript presented in an intelligible fashion and written in standard English?

Reviewer #1: Yes

Reviewer #4: No

6. Review Comments to the Author

Reviewer #1: Thank you for your work on this submission. I'm not able to perform statistical analysis which I leave to you and the other reviewer(s).

Reviewer #4: General comments: While the overall message is able to be conveyed, much of the wording is confusing and frequently grammatically incorrect. I encountered numerous errors in structure and grammar throughout the paper some of which have been highlighted in the specific comments section; however, there are many more throughout. Suggest review by another editor or thorough reworking to refine the syntax.

Was there a power analysis? Should include this in the statistical analysis section, especially given the small numbers for the non-TL groups, which may make the findings for these groups irrelevant. While this is addressed briefly in the drawbacks section, it should also be addressed in the discussion section and backed up by literature where applicable.

Lines 217 to 270 reads like a review of sagittal parameters and, while helpful, detracts from the overall findings of the study and is – to some degree – extraneous. If the purpose of the study is to describe an improvement in sagittal balance after PKP, then the discussion section should be focused on discussing these findings with appropriate references to support these findings (which the authors do in the latter half of the discussion section.) The discussion section needs to be trimmed down and focused on the relevant results.

Specific Comments

Line 25: Reword this sentence – would not use contractions in a scientific paper. i.e. “However, the influence of kyphoplasty on spino-pelvic aligment and global sagittal balance when performed at specific treatment sites in the spine remains unclear.”

Line 37: Remove “of;” Sentence beginning

Line 46: improvement of what in the TL group? Needs to be reworded.

Line 69: Needs references to back up this claim, possibly naming specific examples.

Line 72: Please combine with another paragraph – this sentence should not stand alone as its own paragraph.

Line 76: Sentence beginning “Whether the effect…” is confusingly worded.

Line 101: Be specific regarding “hip and knee joint limitations.” I see that you addressed this in the previous reviewer comments, but it bears explaining in the text. Otherwise it remains confusing and vague.

Line 224: Difficult to understand this sentence as written.

Line 238: Please provide a reference for this paragraph

Line 327: The sentence “After PKP, back pain…” is conjecture. The study did not measure back muscle strength and cannot claim improvement in strength as a reason for improvement in sagittal balance.

7. PLOS authors have the option to publish the peer review history of their article (what does this mean?). If published, this will include your full peer review and any attached files.

Reviewer #1: No

Reviewer #4: No

---

## [Author Response · Author response to Decision Letter 1]

28 Dec 2019

Reviewer's Responses to Questions：

1. If the authors have adequately addressed your comments raised in a previous round of review and you feel that this manuscript is now acceptable for publication, you may indicate that here to bypass the “Comments to the Author” section, enter your conflict of interest statement in the “Confidential to Editor” section, and submit your "Accept" recommendation.

Reviewer #1: All comments have been addressed

Reviewer #4: (No Response)

Response: Thanks for the reviewer's affirmation of our revised manuscript. We revised the manuscript again according to the reviewer's comments, and strived to meet PLOS ONE’s publication criteria.

 2. Is the manuscript technically sound, and do the data support the conclusions?

Reviewer #1: Yes

Reviewer #4: Partly

Response: We appreciate the reviewer’s comments. We asked professional statisticians to perform a statistical analysis of our data again, and added power analysis at the recommendation of reviewers.

3. Has the statistical analysis been performed appropriately and rigorously? 

Reviewer #1: I Don't Know

Reviewer #4: Yes

Response: Thanks to the reviewers for their affirmation of our statistical methods. All of our data statistics programs are consulted by statistical professionals, and it is believed that the statistical methods applied should be appropriate and rigorous.

4. Have the authors made all data underlying the findings in their manuscript fully available?

Reviewer #1: Yes

Reviewer #4: Yes

Response: Thanks for the two reviewers. All the data involved in our manuscript has been uploaded as an attachment during the submission. 

5. Is the manuscript presented in an intelligible fashion and written in standard English?

Reviewer #1: Yes

Reviewer #4: No

Response: We invited native English speaking professionals to proofread our manuscript and revised the grammar and writing issues in the manuscript again. Please review it again and give us your valuable comments. Thank you!

6. Review Comments to the Author

Reviewer #1: Thank you for your work on this submission. I'm not able to perform statistical analysis which I leave to you and the other reviewer(s). 

Responses to Reviewer #1

We would like to extend a special thank you for your helpful comments and recognition of our revised manuscript.

Responses to Reviewer #4:

We would like to extend a special thank you for your helpful comments and constructive criticism.

1) Comment：General comments: While the overall message is able to be conveyed, much of the wording is confusing and frequently grammatically incorrect. I encountered numerous errors in structure and grammar throughout the paper some of which have been highlighted in the specific comments section; however, there are many more throughout. Suggest review by another editor or thorough reworking to refine the syntax.

Response: Thanks for your valuable advice. We are very sorry for the many grammar and writing problems in the manuscript. We consulted native English speaking professionals to review our manuscript and made corresponding modifications to the grammar and writing issues in the text. Please review it again and give us your valuable comments. Thank you!

2) Comment：Was there a power analysis? Should include this in the statistical analysis section, especially given the small numbers for the non-TL groups, which may make the findings for these groups irrelevant. While this is addressed briefly in the drawbacks section, it should also be addressed in the discussion section and backed up by literature where applicable.

Response: We appreciate the reviewer’s comments. We have supplemented power analysis in the statistical analysis section (Line221-227) and discussed the results of power analysis in the discussion section (Line331-340). The power of the study was estimated using a post-hoc analysis with G*Power software (version 3.1.9.4, Franz Faul, Universität Kiel, Germany).With effect size of 0.8 and 0.05 level of statistical significance, the actual power of TL group was calculated to be 0.99. However, the powers of non-TL groups were ranged from 0.55 to 0.65. The reason is that osteoporotic vertebral compression fractures mainly occurred in the thoracolumbar region (T10-L2). Therefore, the number of patients in non-TL group was relatively small. And some of these patients also had fractures in the thoracolumbar region at the same time. In order to avoid confounding factors in the statistical analysis, we excluded patients with fractures in two or more different regions. Therefore, the number of patients in non-TL group was even smaller, which may have resulted in a higher statistical bias. 

3) Comment：Lines 217 to 270 reads like a review of sagittal parameters and, while helpful, detracts from the overall findings of the study and is – to some degree – extraneous. If the purpose of the study is to describe an improvement in sagittal balance after PKP, then the discussion section should be focused on discussing these findings with appropriate references to support these findings (which the authors do in the latter half of the discussion section.) The discussion section needs to be trimmed down and focused on the relevant results.

Response: Thank you for your valuable comments. According to your advice, we have reduced some parts of the discussion. 

 In most cases, global sagittal balance is usually determined by measuring the SVA. Lines 243 to 271 explained why we chose SSA, TPA instead of SVA as the main parameters. We consider that this part should be kept in the manuscript. Thank you!

4) Comment：Specific Comments 

Line 25: Reword this sentence – would not use contractions in a scientific paper. i.e. “However, the influence of kyphoplasty on spino-pelvic aligment and global sagittal balance when performed at specific treatment sites in the spine remains unclear.”

Line 37: Remove “of;” Sentence beginning

Line 72: Please combine with another paragraph – this sentence should not stand alone as its own paragraph.

Response: We are very sorry for these mistakes. We have made corresponding modifications according to the reviewer’s comments. Thank you! 

5) Comment：

Line 46: improvement of what in the TL group? Needs to be reworded.

Response: Thank you for your suggestion. We have changed to “When PKP was performed, the improvement of sagittal balance parameters of TL group was the best in the three groups.” 

6) Comment：

Line 69: Needs references to back up this claim, possibly naming specific examples.

Response: Thank you for your valuable advice. We have added the corresponding references to the manuscript, and taken spinal deformities as an example. 

7) Comment：

Line 76: Sentence beginning “Whether the effect…” is confusingly worded.

Response: Thank you for your valuable comments. We have changed to “Whether there is a difference in the effect of thoracolumbar fracture site on sagittal balance has not yet been studied.” 

8) Comment：

Line 101: Be specific regarding “hip and knee joint limitations.” I see that you addressed this in the previous reviewer comments, but it bears explaining in the text. Otherwise it remains confusing and vague.

Response: Thank you for your valuable comments. We have added an explanatory note to the manuscript to prevent readers from having the same doubts as you. We have changed to “Patients with hip and knee joint limitations (a history of hip and knee joint diseases, or abnormal hip and knee joint mobility in the medical records).”

9) Comment：

Line 224: Difficult to understand this sentence as written.

Line 238: Please provide a reference for this paragraph

Response: Considering that most of the first half of the discussion section is a review of sagittal parameters, we have deleted some paragraphs according to the reviewer’s comments. Line 224 and Line 238 have been removed already.

10) Comment：

Line 327: The sentence “After PKP, back pain…” is conjecture. The study did not measure back muscle strength and cannot claim improvement in strength as a reason for improvement in sagittal balance.

Response: Thank you for your valuable comments. At present, the influence of back muscle strength on sagittal balance is only our conjecture, which is not suitable for the discussion section. We have deleted it from the manuscript. 

Special thanks to you for your good comments.

---

## [Editor Report · Decision Letter 2]

8 Jan 2020

PONE-D-19-23099R2

Percutaneous Kyphoplasty for Osteoporotic Vertebral Compression Fractures Improves Spino-Pelvic Alignment and Global Sagittal Balance Maximally in the Thoracolumbar Region

PLOS ONE

Dear Mr. Sun,

Thank you for submitting your manuscript to PLOS ONE. After careful consideration, we feel that it has merit but does not fully meet PLOS ONE’s publication criteria as it currently stands. Therefore, we invite you to submit a revised version of the manuscript that addresses the points raised during the review process.

The authors have responded well by addressing all the reviewers' comments. However, I regret having to return the manuscript for one very minor correction. This is because no edits can be performed after acceptance.

The new paragraph on power analysis needs some improvement. It currently says:

"With effect size of 0.8 and 0.05 level of statistical significance, the actual power of TL group was calculated to be 0.99. Thus, our sample size of 71 patients was adequate, and the TL group was sufficiently powered to detect the effect of PKP on sagittal balance in patients with OVCF.

However, the powers of non-TL groups were ranged from 0.55 to 0.65. Based on these findings, the evidence is insufficient to investigate the parameters in non TL groups. Thus, additional studies with high-quality and large-scale are still warranted. "

It should say:

"With effect size of 0.8 and 0.05 level of statistical significance, the TL group (n=71) achieved a power of 0.99. Thus, the TL group was sufficiently powered to detect the effect of PKP on sagittal balance in patients with OVCF. However, power in the non-TL groups ranged from 0.55 to 0.65. Based on these findings, there is insufficient data to investigate the parameters in the non-TL groups. Thus, additional studies with proper large-scale cohorts are still warranted."

We would appreciate receiving your revised manuscript by Feb 22 2020 11:59PM. To enhance the reproducibility of your results, we recommend that if applicable you deposit your laboratory protocols in protocols.io, where a protocol can be assigned its own identifier (DOI) such that it can be cited independently in the future. For instructions see: http://journals.plos.org/plosone/s/submission-guidelines#loc-laboratory-protocols

We look forward to receiving your revised manuscript.

Kind regards,

Alejandro A. Espinoza Orías, PhD

Academic Editor

PLOS ONE

---

## [Author Response · Author response to Decision Letter 2]

11 Jan 2020

Comment：

The new paragraph on power analysis needs some improvement. It currently says:

"With effect size of 0.8 and 0.05 level of statistical significance, the actual power of TL group was calculated to be 0.99. Thus, our sample size of 71 patients was adequate, and the TL group was sufficiently powered to detect the effect of PKP on sagittal balance in patients with OVCF.

However, the powers of non-TL groups were ranged from 0.55 to 0.65. Based on these findings, the evidence is insufficient to investigate the parameters in non TL groups. Thus, additional studies with high-quality and large-scale are still warranted. "

It should say:

"With effect size of 0.8 and 0.05 level of statistical significance, the TL group (n=71) achieved a power of 0.99. Thus, the TL group was sufficiently powered to detect the effect of PKP on sagittal balance in patients with OVCF. However, power in the non-TL groups ranged from 0.55 to 0.65. Based on these findings, there is insufficient data to investigate the parameters in the non-TL groups. Thus, additional studies with proper large-scale cohorts are still warranted."

Response: We would like to extend a special thank you for your helpful comments and constructive criticism. According to your advice, we have made corresponding modifications in the manuscript. Thank you!

---

## [Editor Report · Decision Letter 3]

14 Jan 2020

Percutaneous kyphoplasty for osteoporotic vertebral compression fractures improves spino-pelvic alignment and global sagittal balance maximally in the thoracolumbar region

PONE-D-19-23099R3

Dear Dr. Sun,

We are pleased to inform you that your manuscript has been judged scientifically suitable for publication and will be formally accepted for publication once it complies with all outstanding technical requirements.

With kind regards,

Alejandro A. Espinoza Orías, PhD

Academic Editor

PLOS ONE
---

## [Editor Report · Acceptance letter]

23 Jan 2020

PONE-D-19-23099R3 

Percutaneous kyphoplasty for osteoporotic vertebral compression fractures improves spino-pelvic alignment and global sagittal balance maximally in the thoracolumbar region 

Dear Dr. Sun:

I am pleased to inform you that your manuscript has been deemed suitable for publication in PLOS ONE. Congratulations! Your manuscript is now with our production department. 

With kind regards,

on behalf of

Dr. Alejandro A. Espinoza Orías 

Academic Editor

PLOS ONE